# Bridge the Gap Between Architecture Spaces via A Cross-Domain Predictor

**Yuqiao Liu**[*1]**, Yehui Tang**[*2,3]**, Zeqiong Lv**[1]**, Yunhe Wang**[3]**, Yanan Sun**[†1]
[1]College of Computer Science, Sichuan University
[2]School of Artificial Intelligence, Peking University    [3]Huawei Noah's Ark Lab
lyqguitar@gmail.com, yhtang@pku.edu.cn, lzq_june@foxmail.com,
yunhe.wang@huawei.com, ysun@scu.edu.cn

## Abstract

Neural Architecture Search (NAS) can automatically design promising neural architectures without artificial experience. Though it achieves great success, prohibitively high search cost is required to find a high-performance architecture, which blocks its practical implementation. Neural predictor can directly evaluate the performance of neural networks based on their architectures and thereby save much budget. However, existing neural predictors require substantial annotated architectures trained from scratch, which still consume many computational resources. To solve this issue, we propose a Cross-Domain Predictor (CDP), which is trained based on the existing NAS benchmark datasets (*e.g.*, NAS-Bench-101), but can be used to find high-performance architectures in large-scale search spaces. Particularly, we propose a progressive subspace adaptation strategy to address the domain discrepancy between the source architecture space and the target space. Considering the large difference between two architecture spaces, an assistant space is developed to smooth the transfer process. Compared with existing NAS methods, the proposed CDP is much more efficient. For example, CDP only requires the search cost of 0.1 GPU Days to find architectures with 76.9% top-1 accuracy on ImageNet and 97.51% on CIFAR-10. The source code will be available [3].

## 1   Introduction

Neural Architecture Search (NAS) [16] can automatically design promising architectures of deep neural networks for real-world applications, such as image classification, language modeling, and medical image segmentation [50, 46, 62]. In recent years, there have been many architectures searched by NAS methods, which can surpass the performance of those manually designed [43, 23]. Generally, performing NAS methods often requires thousands of GPU Days. Unfortunately, such a large amount of computing resources is not necessarily available to most researchers and enterprises interested. As a result, NAS researchers are paying increasing attention to improve the efficiency of NAS through developing effective acceleration methods, to complete the search within an acceptable cost [34].

Generally speaking, most of the existing search acceleration methods are developed based on some heuristic strategies, *e.g.*, reducing the size of search space [65, 15], searching upon a smaller proxy dataset [43, 33], and sharing the parameters of different architectures (weight sharing) [44, 42].

---

[*]Equal contribution.

[†]Corresponding author.

[3]https://github.com/lyq998/CDP (Pytorch)
  https://gitee.com/mindspore/models/tree/master/research/cv/CDP (MindSpore)

36th Conference on Neural Information Processing Systems (NeurIPS 2022).

However, the shortcomings of the above methods are obvious. Reducing the size of search space also limits the diversity of architectures [52], and the architectures optimized on the proxy dataset are not guaranteed to perform best on the target dataset [4]. Weight sharing first trains a supergraph that is manually designed in advance and then searches subgraphs as the potential architectures in the supergraph. The weights in the same components between different architectures are shared from the trained supergraph without further training, thus the efficiency of performing of NAS is greatly improved. However, the weight sharing also suffers from performance instability and dramatic performance collapse [8, 6], which severely limits its application.

Neural predictor is a promising acceleration method, which directly estimates the performance of neural architectures without any training of the neural network during NAS [55]. For example, Sun *et al.* [49] proposed an end-to-end performance predictor that employed random forest [21] which reduced the consumed time of NAS significantly. Tang *et al.* [54] used a graph convolution network (GCN) as the neural predictor and fully exploited the unlabeled architectures to improve the prediction accuracy. Though these predictor-based methods tend to be more efficient than other methods, sufficient architectures which play as the annotated examples to train the predictor, are still required to sample from the target search space. These sampled architectures are usually required to train fully to obtain their actual performance [55, 49]. A large search space requires a mass of samples, which becomes a bottleneck for further improving predictor-based methods. Consequently, neural predictor still requires massive computing costs largely due to annotating the sampled architectures.

Fortunately, we note that many benchmark datasets have been released for NAS research, such as NAS-Bench-101 [60], NAS-Bench-201 [13], NAS-Bench-nlp [29], NAS-Bench-ASR [38], *etc.* These datasets contain a great number of architectures with known real performance, and constructing them has consumed massive computational resources. If these data can come in handy, there is no need to annotate other architectures for predictors and the search cost is greatly reduced.

However, the search spaces of these datasets are small and have a huge difference from the large search spaces adopted by popular NAS algorithms such as DARTS [33] and ProxylessNAS [4]. Although the architectures are all made up of operations, the settings are quite different, such as the types and the maximum number of operations. As a result, these well-trained benchmark architectures cannot be directly used to build predictors.

In this paper, we develop a Cross-Domain Predictor (dubbed as CDP) by exploiting the existing NAS benchmarks, and then the predictor is implemented to find high-performance neural architectures in large search spaces.

The contributions of CDP are summarized as follows: **1)** We divide the architecture space for subspace adaptation, and propose a progressive subspace partition to accurately exploit the local information. **2)** We design an assistant space, to achieve smoother transfer from the source space to the target one because of the significant differences between searching domains. **3)** We experimentally show that CDP can search promising architectures with only 0.1 GPU days on ImageNet and CIFAR-10, achieving 76.9% and 97.51% classification accuracy, respectively.

## 2   Related Works

### 2.1   NAS and Neural Predictor

In general, there are three important parts in NAS [16], namely architecture space[1], search strategy, and performance evaluation. During the search process, each NAS method first specifies the architecture space that includes all the potential architectures to be searched. Secondly, a particular search strategy is utilized to perform the search. In practice, reinforcement learning [26], evolutionary computation [1], and gradient-based methods are the mainstream search strate-

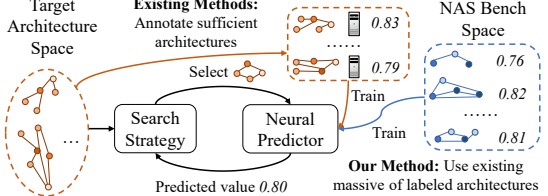

Figure 1: Neural predictor-assisted NAS. Existing neural predictors need to annotate sufficient architectures from the target space, which incurs a high computational cost. Our method uses existing NAS benchmark datasets where massive available labeled architectures can be used to train the neural predictor.

---

[1]It is also named as search space in some literature, in this paper, we treat them as the same.

gies widely used. Finally, the performance evaluation assesses the architectures searched to guide the search strategy.

In recent years, how to accelerate the performance evaluation is one of the most valuable research directions in NAS [34]. As a promising acceleration method, neural predictors have received great attention from researchers since 2017 [11]. The red and black parts in Fig. 1 illustrate a typical neural predictor-assisted NAS for a glance. Recently, with the design of new predictors [49, 54, 55], the prediction performance of neural predictors has been continuously improved. However, existing predictors request the source and the target architecture spaces to be the same. If the predictor makes predictions for a specific NAS method, it must collect sufficient labeled architectures from the search space of the corresponding NAS method in advance. The CDP algorithm we proposed in this paper gets rid of the constraint that the source and the target architecture spaces must be the same. Particularly, CDP directly utilizes the existing benchmark datasets as the training architectures in the source space, yet to predict the architectures in other target spaces. In this way, the neural predictor will not need to sample labeled data from the target space of the particular NAS method, thus greatly reducing the cost of annotating the training architectures.

## 2.2 Transfer Learning and Domain Adaptation

Transfer learning is to apply knowledge learned in previous domains to other domains, thus reducing the need and effort to recollect labeled data [40]. Specifically, domain adaptation is a subcategory of transfer learning, where the source and target tasks are the same and labeled data is available in the source domain, which is the scenario of CDP. According to [64], transfer learning can be classified into two different categories based on the consistency between the source and the target domains. If the feature spaces are different in both domains, the scenario is termed heterogeneous transfer learning. Otherwise, it is homogeneous transfer learning. The proposed CDP algorithm in this paper follows heterogeneous transfer learning because the architecture spaces are different. But observing that much more current literature addresses the scenario of homogeneous transfer learning [10], we proposed a uniform encoding strategy that unifies the architecture spaces, so that it can be treated as homogeneous transfer learning.

The feature-based transfer is one of the most prevalent methods. It aims at reducing the distribution difference between the source domain and the target domain with the help of some instances. A common metric for measuring the difference is Maximum Mean Discrepancy (MMD) [18], which can quantify the difference by calculating the average distance of the instances sampled from both domains [64]. Inspired by MMD, many variants and improved approaches are proposed such as MK-MMD [19], weighted MMD [59], and Local Maximum Mean Discrepancy (LMMD) [63]. Among them, LMMD can take advantage of local affinity for more delicate adaptation and can achieve extraordinary results. Specifically, LMMD first classifies the domain into several subspaces and then aligns the subspaces of the same category in pairs. Secondly, LMMD uses the pseudo-labels output by the neural network to classify the target data throughout the training process. However, it is well known that the neural network cannot achieve satisfactory results in the initial training stage, that is, the pseudo-label obtained through it may be seriously inconsistent with the real label in the early training process. In this paper, we propose a progressive approach to alleviate the disadvantages of misclassification.

# 3 Approach

In this section, we will document how the proposed cross-domain predictor bridges the gap between different architecture spaces. The overall process of the proposed CDP algorithm is shown in Fig. 2.

## 3.1 Formulation

Supposing the architectures in the search space $\mathcal{X}$ are $\mathbf{X} = \{\mathbf{x}_1, \mathbf{x}_2, ..., \mathbf{x}_n\}$, and their respective performances in space $\mathcal{Y}$ are $\mathbf{Y} = \{y_1, y_2, ..., y_n\}$, the predictor $P : \mathcal{X} \rightarrow \mathcal{Y}$, which is trained to predict the performance of architecture, can be formulized by Equation (1):

$$\min_W \frac{1}{n} \sum_{n=1}^{n} \mathcal{L}(P(W, \mathbf{x}_n), y_n), \tag{1}$$

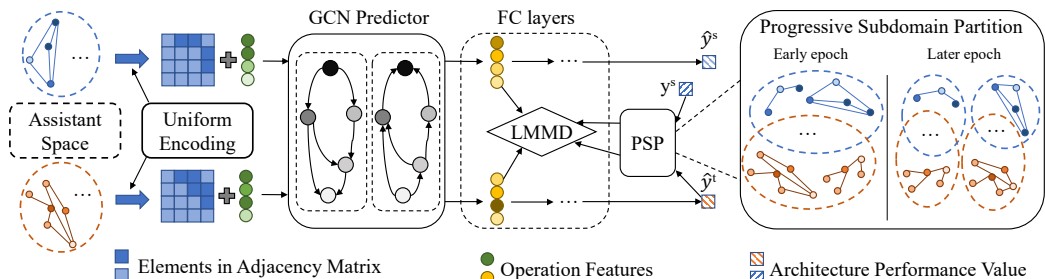

Figure 2: The CDP flowchart for evaluating neural architectures. The blue and red dashed circles represent the source architecture space and target architecture spaces, respectively. The size of assistant space is between those of source and target spaces, and this space is used in early training for a smoother transfer. Architectures from these spaces are required to go through the uniform encoding to be represented by adjacency matrix $M$ and operation features $O_0$. Note that the true label of source architecture data and the pseudo-label of target architecture data, *i.e.*, $y^s$ and $\hat{y}^t$, need to go through the proposed Progressive Subspace Partition (PSP) strategy first, and then are used to calculate LMMD. The detailed figure of PSP is displayed on the right. In the beginning, this strategy divides all architectures into the same space. As the neural network training progresses, the number of subspaces will continue to increase until it finally reaches the specified maximum number. The partition of subspace is based on true and pseudo labels.

where $W$ denotes the trainable weight of predictor $P$, and $\mathcal{L}$ is the loss function. After the training, the predictor with trained weight $W$ will evaluate other architectures, say $\mathbf{x}_i$ where $i$ is not in $\{1, 2, ..., n\}$, via Equation (2):

$$\hat{y}_i = P(W, \mathbf{x}_i). \tag{2}$$

Mathematically, neural architectures can be presented as directed graphs, and a common encoding scheme of architecture $\mathbf{x}$ is to use feature vectors of operations $O_0$ and adjacency matrix $M$, which are further collectively employed to represent the type of each operation and the connection between operations. GCN [28] is a prevalent technique tackling non-Euclidean data such as graphs and has achieved great success in representing neural architectures [55]. Therefore, it is used as the first component in $P$ to extract features of neural architecture $G : \mathcal{X} \to \mathcal{Z}$, where $\mathcal{Z}$ is the feature space.

In this paper, we follow the convention and choose the GCN version implemented in [55] as the neural predictor, and the feature vectors are updated by Equation (3):

$$O_{l+1} = \frac{1}{2}\text{ReLU}(MO_lW_l^+) + \frac{1}{2}\text{ReLU}(M^TO_lW_l^-), \tag{3}$$

where $M^T$ is the transposed matrix of $M$ and ReLU is an activation function. $W_l^+$ and $W_l^-$ are two trainable weights of the $l$-th layer in GCN. Specifically, the last two terms of Equation (3) allow the graph information to flow forward and backward, respectively, which makes it more suitable for dealing with directed graphs like neural architectures. In addition, the second component of $P$ is a hypothesis $h : \mathcal{Z} \to \mathcal{Y}$, consisting of several fully connected layers, and is tailed to the GCN for obtaining the predicted performance values. As for loss function $\mathcal{L}$, we also follow the convention suggested in the reference [55] using mean square error.

## 3.2 Progressive Subspace Adaptation

Although neural predictor can largely save search cost for NAS, the limitation of itself is also serious: it cannot be well trained due to the lack of available labeled architectures of the corresponding NAS algorithm. Observing that many architecture datasets have been released, these data can be potentially used to train the neural predictors. However, the distribution of these massive labeled architectures is significantly different from that of the architectures from the popular search space that is often large. In principle, domain adaptation is useful for reducing the distribution gap between these architecture spaces. Let $\widetilde{\mathcal{D}}_S$ and $\widetilde{\mathcal{D}}_T$ be the source and target distributions over $\mathcal{Z}$. We will specifically introduce how to bridge the gap between $\widetilde{\mathcal{D}}_S$ and $\widetilde{\mathcal{D}}_T$ in the proposed method as below.

Research has shown that deep neural networks have the ability to learn the transferable features well [61, 39]. In addition, many methods [35, 63] add metrics of discrepancy to the objective function

to jointly optimize the neural network. Based on Equation (1), the term of adaptation regularizer $d(\cdot, \cdot)$ is added, and the objective function becomes:

$$\min_{W} \frac{1}{n^s} \sum_{n=1}^{n^s} \mathcal{L}(P(W, \mathbf{x}_n^s), y_n^s) + \theta d(\widetilde{\mathcal{D}}_S, \widetilde{\mathcal{D}}_T), \tag{4}$$

where $\mathbf{x}_n^s$ and $y_n^s$ denote architectures with labels in source search space $\mathcal{X}_S$, and $\theta > 0$ is a penalty parameter.

Maximum Mean Discrepancy (MMD) [18] is a classic metric and has been widely used in domain adaptation. Let $\mathcal{H}_k$ denote the reproducing kernel Hilbert space (RKHS) endowed with a characteristic kernel $k$, and $\widetilde{\mathcal{U}}_S, \widetilde{\mathcal{U}}_T$ are samples of sizes $n_s, n_t$ drawn from $\widetilde{\mathcal{D}}_S, \widetilde{\mathcal{D}}_T$ respectively. The regularizer term $d(\cdot, \cdot)$ can be implemented by MMD, and the unbiased estimator of MMD is defined by latent representation $\mathbf{z}^s, \mathbf{z}^t$:

$$\hat{d}_k(\widetilde{\mathcal{D}}_S, \widetilde{\mathcal{D}}_T) \triangleq \|\frac{1}{n_s} \sum_{\mathbf{z}_i^s \in \widetilde{\mathcal{U}}_S} \phi(\mathbf{z}_i^s) - \frac{1}{n_t} \sum_{\mathbf{z}_j^t \in \widetilde{\mathcal{U}}_T} \phi(\mathbf{z}_j^t)\|_{\mathcal{H}_k}^2, \tag{5}$$

where $\phi$ denotes feature map function mapping original space to $\mathcal{H}_k$ and it is associated with $k$, $k(\mathbf{z}^s, \mathbf{z}^t) = \langle \phi(\mathbf{z}^s), \phi(\mathbf{z}^t) \rangle$ where $\langle \cdot, \cdot \rangle$ means the inner product operation.

Considering that the differences between search spaces are quite large, the gap cannot be handled well if only MMD is used. In addition, MMD is a global domain adaptation, which often ignores the local relationship. In order to take full use of the local information between architectures, we present a progressive subspace adaptation strategy. First, the source space and the target space are divided into several subspaces whose number will increase during the training process. Secondly, MMD loss is used between corresponding subspaces for adaptation. Specifically, these two spaces will be divided according to the performance $\mathbf{Y} = \{y_1, y_2, ..., y_n\}$ of architectures. Between these two spaces, the architectures with good performance may share some characteristics, such as long shortcut connections and preferred operations.

We directly use the true labels $y$ of architectures in the sources space and predict the pseudo-labels $\hat{y}$ of architectures in the target space to divide the subspaces. The performance value $y_i$ is a scalar, which needs to be classified first to become a classification label $\mathbf{y}_i$ that is a vector. Secondly, the weight $w_i^c$ which presents the $x_i$ belonging relationship to the $c$-th subspace can be calculated via Equation (6):

$$w_i^c = \frac{y_i^c}{\sum_{\mathbf{y}_j \in \mathbf{Y}} y_j^c}, \tag{6}$$

where $y_i^c$ denotes the $c$-th element of label vector $\mathbf{y}_i$. With the weight $w_i^c$ and Equation (5), the unbiased estimator of the discrepancy between subspaces becomes Equation (7):

$$\hat{d}_k(\widetilde{\mathcal{D}}_S, \widetilde{\mathcal{D}}_T) \triangleq \frac{1}{C} \sum_{c=1}^{C} \|\sum_{\mathbf{z}_i^s \in \widetilde{\mathcal{U}}_S} w_i^{sc} \phi(\mathbf{z}_i^s) - \sum_{\mathbf{z}_j^t \in \widetilde{\mathcal{U}}_T} w_j^{tc} \phi(\mathbf{z}_j^t)\|_{\mathcal{H}_k}^2. \tag{7}$$

This discrepancy is termed Local Maximum Mean Discrepancy (LMMD) [63] that takes the correlation from the same category of subspaces into consideration.

With the help of the kernel $k$, Equation (7) can be further modified to obtain the following formula for easy calculation of LMMD:

$$\begin{aligned} \hat{d}_k(\widetilde{\mathcal{D}}_S, \widetilde{\mathcal{D}}_T) = &\frac{1}{C} \sum_{c=1}^{C} [\sum_{i=1}^{n^s} \sum_{j=1}^{n^s} w_i^{sc} w_j^{sc} k(\mathbf{z}_i^s, \mathbf{z}_j^s) \\ &+ \sum_{i=1}^{n^t} \sum_{j=1}^{n^t} w_i^{tc} w_j^{tc} k(\mathbf{z}_i^t, \mathbf{z}_j^t) - 2 \sum_{i=1}^{n^s} \sum_{j=1}^{n^t} w_i^{sc} w_j^{tc} k(\mathbf{z}_i^s, \mathbf{z}_j^t)]. \end{aligned} \tag{8}$$

However, there is a severe issue that the pseudo-label $\hat{y}$ may be incorrect, especially when the neural network is not fully trained. This will further affect the accuracy of $\hat{d}_k(\widetilde{\mathcal{D}}_S, \widetilde{\mathcal{D}}_T)$ in Equation (8).

In order to alleviate the misclassification due to inaccurate pseudo-labels, we propose a progressive subspace partition strategy. All the architectures are partitioned into one space in the early stage of neural network training so that there will be no classification errors caused by incorrect pseudo labels. As the training progresses, the neural network can predict more accurately. At this time, we slowly increase the number of subspaces. Finally, the number of subspaces will end at the specified maximum $K$.

**Progressive Subspace Partition:** Supposing the neural network is specified to be trained for a total of $E$ epochs, and the maximum number of subspaces is $K$, when the training progresses arrive at the $e$-th epoch, the number of subspace categories $C_e$ can be scheduled by Equation (9):

$$C_e = Sche(e; E, K), \tag{9}$$

and $\forall i < j, Sche(i; E, K) < Sche(j; E, K)$. Based on experience and experiments, in this paper we let $C_e = K - \lfloor cos(\frac{\pi}{2E}e) * K \rfloor$, where $\lfloor \cdot \rfloor$ denotes the round-down operation. $C_e$ calculated by this schedule meets the characteristics discussed above, *i.e.*, when $e$ is relatively small, $C_e$ is equal to 1, while $e$ is close to $E$, $C_e$ will be $K$. Furthermore, we use the cosine function to increase the number of epochs with a small value of $C_e$. This strategy ensures that the neural network has been sufficiently trained before the $C_e$ becomes larger. Therefore, the occurrence of classification errors will be greatly reduced.

After determining the number of subspace categories $C_e$ during the $e$-th training epoch, we can use fractile to generate a new label $\mathbf{y}$ for the original label $y$ which is a scalar. To be more specific, the fractiles $\{f_0, f_1, f_2, ..., f_{C_e}\}$ of scalar labels need to be found at first, where $f_{C_e}$ is the largest one in the labels and $f_0$ is the lower bound. Then, we can find the corresponding subspace category $c$ for $y_i$ if $f_{c-1} < y_i \le f_c$. After that, we set the $c$-th element of $\mathbf{y}_i$ to one and other elements as zeros. Finally, the classification vector $\mathbf{y}_i$ is used by Equation (6) for the calculation.

Here, we will further analyze the role of the proposed progressive subspace partition strategy. When the number of subspaces is one, the numerator $y_i^c$ in Equation (6) is always one, and the denominator $\sum_{\mathbf{y}_j \in \mathbf{Y}} y_j^c$ is equal to the number of source (or target) data. Hence, $w_i^{sc}$ is equal to $1/n^s$ and $w_i^{tc}$ is equal to $1/n^t$. As a result, the unbiased estimator of LMMD (Equation (7)) becomes that of MMD (Equation (5)) which is a global adaptation. While the number of subspaces increases to the specified maximum number $K$, the measurement of the discrepancy we used is actually LMMD which focuses on the relationship between the same categories of subspaces to help the subspace adaptation.

## 3.3 Assistant Architecture Space

Considering that the source architecture space and target architecture space are quite different, it is difficult to directly transfer the source architecture space to the target space. To address this issue, we design an assistant architecture space to make the transfer smoother.

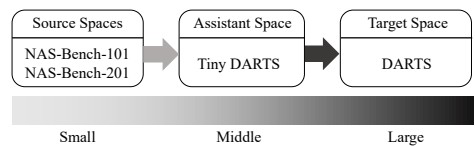

Figure 3: An example of assistant space.

Generally, the source space, where all architectures can be trained from scratch to obtain the performance of each architecture, is often relatively small. However, the target architecture spaces are often large in practice. An example is given based on two widely used architecture spaces, namely NAS-Bench-101 [60] and DARTS [33]. In terms of the number of possible architectures, NAS-Bench-101 is 423K and DARTS is about $10^{18}$. In terms of the maximum number of operations in a cell, NAS-Bench-101 is 5 while DARTS is 8. Thus, the size of the assistant space should be between the size of the source space and the target space. Besides, other features should be similar to these two architecture spaces. In this paper, we specify the assistant space more similar to the target space.

Specifically, an assistant space for NAS-Bench-101, NAS-Bench-201 [13], and DARTS are designed following the above principles, dubbed as 'tiny DARTS' space. Fig. 3 is an illustration of this assistant space, where we use dark black to represent the large architecture space and gray to represent the smaller one. The participation of the assistant space makes the adaptation be a continuous process and alleviates the inadaptability caused by large differences. The only difference between tiny DARTS and DARTS is that the number of intermediate nodes has changed from 4 to 3. This directly reduces the number of operations from 8 to 6 which is equal to the situation in NAS-Bench-201. Except

for this change, any other settings, including the types of operations, *etc.*, have not been changed. Therefore, the features in DARTS have also been preserved. Specifically, there are two input nodes in tiny DARTS, and all intermediate nodes are connected to the output node.

In the early training epochs of CDP, the assistant space (*e.g.*, tiny DARTS) is used as the target domain, *i.e.*, the input at the bottom in Fig. 2. In this stage, CDP focuses on bridging the smaller gap between the source space and the assistant space. After several training epochs, the gap between the source space and the target space can be also reduced as the assistant space and target space are very similar. Therefore, it is much easier to reduce the gap between the source and target spaces at this time. As a result, the space transfer process becomes smoother with the assistant space, which improves the prediction performance.

After the determination of the target domain, an encoding strategy is required for different search spaces. We propose a uniform strategy with four main steps, where The detailed descriptions are shown in Appendix A.

### 3.4 Theoretical Analysis

In this section, we give a theoretical analysis of the expected error of target architecture search space. We first introduce Lemma 1 which is a basic theorem in domain adaptation, and then further deduce the target error bound of CDP.

**Lemma 1.** *[3] Let $\epsilon_T(h)$ and $\epsilon_S(h)$ be the expected error on target and source domain and $\mathcal{H}$ be a hypothesis space, for $h \in \mathcal{H}$:*

$$\epsilon_T(h) \le \epsilon_S(h) + d_{\mathcal{H}}(\widetilde{\mathcal{D}}_S, \widetilde{\mathcal{D}}_T) + \lambda, \tag{10}$$

*where $\lambda = \epsilon_T(h^*) + \epsilon_S(h^*)$ is the combined error of ideal hypothesis $h^* = \arg\min_{h \in \mathcal{H}}(\epsilon_T(h) + \epsilon_S(h))$ on both domains and $d_{\mathcal{H}}(\widetilde{\mathcal{D}}_S, \widetilde{\mathcal{D}}_T)$ is the upper bound of the $\mathcal{A}$-distance [27].*

Let $\widetilde{\mathcal{U}}_{S,train}, \widetilde{\mathcal{U}}_{S,valid}$ be the training and validation datasets randomly sampled from the distribution $\mathcal{D}_S$ respectively, and $\widetilde{\mathcal{U}}_{S,train} \cap \widetilde{\mathcal{U}}_{S,valid} = \emptyset$. After the predictor is trained on the training dataset $\widetilde{\mathcal{U}}_{S,train}$, the validation dataset $\widetilde{\mathcal{U}}_{S,valid}$ is used to validate it. The validation hypothesis $\mathcal{H}'$ is a subset of $\mathcal{H}$, and is only determined by $\widetilde{\mathcal{U}}_{S,train}$. The following theorem provides an upper bound of expected target-domain error in terms of the empirical error on $\widetilde{\mathcal{U}}_{S,valid}$ and MMD $d_k(\widetilde{\mathcal{D}}_S, \widetilde{\mathcal{D}}_T)$.

**Theorem 2.** *Let $d'$ be the VC-dimension of $\mathcal{H}'$, $m$ be the size of $\widetilde{\mathcal{U}}_{S,valid}$, $m'$ be the size of unlabeled samples $\widetilde{\mathcal{U}}_S$ and $\widetilde{\mathcal{U}}_T$. With probability of $1 - \delta$, for $h \in \mathcal{H}'$:*

$$\begin{aligned}
\epsilon_T(h) &\le \hat{\epsilon}_{S,valid}(h) + 2d_k(\widetilde{\mathcal{D}}_S, \widetilde{\mathcal{D}}_T) + \frac{2(d' \log m - \log \delta)}{3m} \\
&+ \sqrt{\frac{2(d' \log m - \log \delta)}{m}} + 4\sqrt{\frac{d' \log(2m') + \log(\frac{4}{\delta})}{m'}} + 2 + \lambda.
\end{aligned} \tag{11}$$

*Proof sketch:* The proof is mainly divided into two steps. We first use the empirical error on source validation dataset $\hat{\epsilon}_{S,valid}(h)$ to represent the upper bound of expected error on source domain $\epsilon_S(h)$:

$$\epsilon_S(h) \le \hat{\epsilon}_{S,valid}(h) + \frac{2(d' \log m - \log \delta)}{3m} + \sqrt{\frac{2(d' \log m - \log \delta)}{m}}. \tag{12}$$

The detailed proof can be found in Equations (19)-(24) in Appendix B. Secondly, the upper bound of the distance $d_{\mathcal{H}}(\widetilde{\mathcal{D}}_S, \widetilde{\mathcal{D}}_T)$ is constrained by MMD $d_k(\widetilde{\mathcal{D}}_S, \widetilde{\mathcal{D}}_T)$, *i.e.*:

$$d_{\mathcal{H}}(\widetilde{\mathcal{D}}_S, \widetilde{\mathcal{D}}_T) \le 2 + 2d_k(\widetilde{\mathcal{D}}_S, \widetilde{\mathcal{D}}_T) + 4\sqrt{\frac{d' \log(2m') + \log(\frac{4}{\delta})}{m'}}, \tag{13}$$

and Equations (25)-(26) in Appendix B show the proof in detail. Finally, by applying these inequalities to Lemma 1, the theorem can be obtained.

Theorem 2 shows that the expected error on the target domain is bound by two terms, the empirical validation error in source domain $\hat{\epsilon}_{S,valid}(h)$ and MMD $d_k(\widetilde{\mathcal{D}}_S, \widetilde{\mathcal{D}}_T)$ which is minimized as one term of the objective function. This means that when observing the predictor performs well on the source validation dataset, the expected error on the target domain will also be small.

Table 1: Comparison with state-of-the-art and neural predictor-assisted NAS methods on ImageNet. We report the results with the lowest cost if there are multiple architectures searched in the literature of the compared methods. The symbol '—' denotes that no results are available in original papers.

| Model | Cost (GPU Days) | Top-1 (%) | Top-5 (%) | FLOPs (M) | # Params (M) |
|---|---|---|---|---|---|
| NASNet-A [65] | 2000 | 74.0 | 91.6 | 564 | 5.3 |
| AmoebaNet-A [43] | 3150 | 74.5 | 92.0 | 555 | 5.1 |
| DARTS [33] | 4 | 73.3 | 91.3 | 574 | 4.7 |
| MobileNetV3 [23] | ≈3000 | 75.2 | 92.2 | 219 | 5.4 |
| EfficientNet B0 [53] | ≈3000 | 76.3 | 93.2 | 390 | 5.3 |
| FairDARTS-B [8] | 0.4 | 75.1 | 92.5 | 541 | 4.8 |
| PC-DARTS [58] | 0.1 | 74.9 | 92.2 | 586 | 5.3 |
| MoGA-A [7] | 12 | 75.9 | 92.8 | 304 | 5.1 |
| DropNAS [22] | 0.6 | 75.5 | 92.6 | 572 | 5.2 |
| DOTS [20] | 0.2 | 75.7 | 92.6 | 581 | 5.2 |
| NAO [37] | 200 | 74.3 | 91.8 | 584 | 11.35 |
| Neural Predictor [55] | 119 Archs[*] | 74.7[**] | — | — | — |
| SemiNAS [36] | 4 | 76.5 | 93.2 | 599 | 6.3 |
| WeakNAS [56] | 2.5 | 76.5 | 93.2 | 591 | 5.5 |
| CDP (Ours) | **0.1** | **76.9** | 93.0 | 548 | 5.4 |

[*] The search cost is 119 architectures trained on ImageNet, not measured in GPU Days.
[**] An approximate value estimated from the figure in its original paper.

# 4 Experiments

In this section, we will empirically investigate the effectiveness of the proposed method, including the experiments conducted on ImageNet [12] and CIFAR-10 [30], and the ablation studies. Our experiments are conducted with PyTorch [41] and MindSpore [25].

## 4.1 Experimental Settings

**Architecture spaces for CDP:** NAS-Bench-101 and NAS-Bench-201 are chosen as the source spaces and DARTS is the target space. The training labels in NAS-Bench-101 and NAS-Bench-201 are normalized separately. This can alleviate the problems caused by different performance scales.

**Search strategy:** Following the convention in neural predictor [55], a large set of architectures in the target search space are randomly sampled and their performance is predicted. In addition, according to the previous work on DARTS search space [5], we limit the max number of skip connections to 2. After that, we choose the architecture with the best-predicted performance as the searched architecture.

## 4.2 Search Results on ImageNet and CIFAR-10

The experimental results of ImageNet are shown in Table 1, where the competitors are all the state-of-the-art NAS methods (shown in the second row) and neural predictor-assisted NAS methods (shown in the third row). As can be seen from Table 1, the proposed CDP method only consumed 0.1 GPU Days to complete the search, and it is one of the least costly search methods. Specifically, compared with the chosen neural predictor-assisted NAS methods, the search cost of CDP is **2000**× less than NAO, **40**× less than SemiNAS, and **25**× less than Weak-NAS. The main reason for the extremely low cost of CDP lies in that CDP can make full use of a large amount of existing data from

Table 2: Comparison with state-of-the-art NAS methods on CIFAR-10. GPU Days is used to measure search cost. 'Acc.' denotes the accuracy and 'Params' denotes the parameters.

| Model | Cost | Acc. (%) | Params (M) |
|---|---|---|---|
| NASNet-A | 2000 | 97.35 | 3.3 |
| ENAS | 0.5 | 97.11 | 4.6 |
| FairDARTS | 0.4 | 97.46 | 2.8 |
| P-DARTS | 0.3 | 97.50 | 3.4 |
| PC-DARTS | 0.1 | 97.43 | 3.6 |
| PVLL-NAS | 0.2 | 97.30 | 3.3 |
| AmoebaNet-A | 3150 | 96.66 ± 0.06 | 3.2 |
| DARTS | 4 | 97.24 ± 0.09 | 3.3 |
| BRP-NAS | 6 | 97.29 ± 0.07 | — |
| CDP (Ours) | **0.1** | **97.51** 97.37 ± **0.0**8 | 3.3 |

the benchmark datasets. In addition, although CDP consumes a similar budget to that of PC-DARTS which is the state-of-the-art NAS method using weight sharing, CDP achieves the best top-1 accuracy. The searched architectures are displayed in Appendix C.

Table 3: The impact of different progressive strategy (PS).

| PS | KTau |
|---|---|
| None | 0.3650 |
| Linear | 0.4949 |
| Cosine | 0.5176 |

Table 4: Ablation study on assistant space (AS).

| AS | Size | KTau |
|---|---|---|
| ✗ | — | 0.5176 |
| ✓ | Small | 0.5051 |
| ✓ | Medium | 0.5306 |

Table 5: Ablation study on domain adaptation approaches.

| Approach | KTau |
|---|---|
| DANN [17] | 0.4686 |
| CORAL [48] | 0.4306 |
| MMD [35] | 0.4549 |
| LMMD [63] | 0.4969 |
| LMMD + PSP | 0.5306 |

The comparison results of CIFAR-10 are shown in Table 2. In this experiment, in addition to the methods compared in Table 1, we also compare with three other state-of-the-art methods. They are ENAS [42], P-DARTS [5], PVLL-NAS [31], and a predictor-assisted NAS method BRP-NAS [14]. The parameters of all comparison methods are similar. In terms of the search cost, CDP requires only 0.1 GPU Days, which is one of the lowest cost methods in the table. In addition, the architecture searched by CDP also outperforms all the NAS methods compared.

## 4.3 Ablation Study

To the best of our knowledge, CDP is the first work of neural predictors designed from the cross-domain view, by directly exploring the architectures of existing NAS benchmark datasets. Therefore, it is not practical to conduct any experiments on NAS-Bench-101 and NAS-Bench-201 by tradition. To deal with it, we create a new dataset for neural predictor termed shallow DARTS by randomly sampling 100 shallow architectures from DARTS search space. In order to save training cost, CIFAR-10 is chosen as the dataset, and these architectures have only 8 layers which are shallower than the architectures reported in Table 2. Moreover, we train these shallow architectures for 50 epochs to further save cost.

After CDP has been trained on NAS-Bench-101 and NAS-Bench-201, it is used to predict the performance of 100 architectures in the shallow DARTS dataset. Following the conventions, we also pay more attention to the relative ranking between the architectures. Therefore, we use Kendall's Tau (KTau) [45] which is widely used to measure the quality of neural predictors [49, 54] in this ablation study. KTau concerns the relative ranking between the architectures, and its values vary in $[-1, 1]$. The larger the value, the more the predicted ranking matches the real ranking.

Table 3 displays the results with and without the proposed progressive subspace partition strategy. Specifically, we specify that the numbers of subspaces are all 3 in the end. 'None' denotes no progressive strategy is used. In addition, progressive subspace partition with the linear increasing rate in Equation (9) is also tested to show the effectiveness of the cosine function. Generally, the performance of the predictor is much better when working with the proposed progressive subspace partition strategy. Furthermore, the cosine function also works better than the linear increasing rate. In summary, the ablation study proves the effectiveness of the proposed progressive subspace partition.

The results of the ablation study on the assistant space are reported in Table 4. The KTau value of the first line is obtained with the help of the progressive subspace partition. On this basis, two assistant spaces are added separately. The first one is similar to DARTS and the number of internal nodes is set as 2, which makes its size be smaller than both the source and target architecture spaces. The second is the tiny DARTS whose size is between the source and target architecture spaces. As can be seen from Table 4, the small assistant space fails to improve the performance of the predictor. But with the help of tiny DARTS (the medium space), the prediction performance can be further improved on the basis of the progressive subspace partition.

Finally, we conduct an ablation study on domain adaptation approaches. In addition to the MMD and LMMD introduced above, domain-adversarial neural network (DANN) [17] and CORrelation ALignment (CORAL) [48] are also compared. We fine-tune the hyper-parameters of these approaches (if feasible), and report the best KTau values they achieved in Table 5. It is obvious that our approach is far better than peer competitors.

To sum up, the ablation studies prove that the proposed progressive subspace partition and assistant space can significantly improve the prediction performance of CDP. Other ablation studies can be found in Appendix E.

# 5   Conclusion

This paper aims to develop a cross-domain predictor, named CDP, which does not require training neural architectures in the particular search space. This goal has been achieved by the two proposed components. Specifically, the progressive subspace partition has been proposed to largely alleviate the issue of inaccurate pseudo-label. In addition, the assistant space has been developed to address the inadaptability caused by large differences between source and target domain. The training architectures of CDP are collected from the existing NAS benchmark datasets, so it does not have to spend a lot of computation cost on training neural architectures as the existing neural predictors do. CDP can find high-performance architectures in the large search space (such as DARTS) with 0.1 GPU Days and achieves the classification accuracy of 76.9% and 97.51% on ImageNet and CIFAR-10, respectively.

## Acknowledgments

This work was sponsored in part by Zhejiang Lab (NO.2022PG0AB02), in part by CAAI-Huawei MindSpore Open Fund, and in part by the Science and Technology Innovation Talent Project of Sichuan Province under Grant 2021JDRC0001. We gratefully acknowledge the support of MindSpore, CANN (Compute Architecture for Neural Networks) and Ascend AI Processor used for this research.

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
