# OpenReview forum: "Bridge the Gap Between Architecture Spaces via A Cross-Domain Predictor"
_NeurIPS.cc/2022/Conference — NeurIPS 2022 Accept_

### Official Review · Reviewer_tD8p · 2022-06-17

**Rating:** 6
**Confidence:** 5
**Soundness:** 3 good
**Presentation:** 3 good
**Contribution:** 3 good

**Summary:**

This work targets on leveraging leverage existing NAS Bench to predict out-of-domain architecture’s performance. The author propose to learn a predictor on network's performance, via closing the domain feature gap between source and target architecture space.

**Questions:**

1. Eq. 9, the scheduler for the Subspace Partition, sounds empirical. The authors may consider down-play its contribution, or compare Eq. 9 with other possible schedulers to show its importance.

2. I can understand the importance of the Assistant Architecture Space. However, is there any possible principles in designing it? Otherwise it will hurt the generality of the proposed method.

3. d_H in Eq. 10 is not defined or discussed, or did I miss anything? I suppose it is a kind of distance measure between two domains?

4. I think Thm 2 is too general to be helpful to the main purpose of this (NAS) work. As far as I understand, the most insight from Thm. 2 is that larger m and m’ would give us a lower error bound. But this conclusion is very general and is not leveraged in this work (thus sounds less important).

**Limitations:**

I think in general the target problem and the method of this work is novel. However, some designs in the method (Eq. 9, and the assistant space) may be empirical and may hurt its practical use.

**Strengths And Weaknesses:**

1. The targeted question quite makes sense: to use prepared NAS benchmarks to learn the performance predictor.

2. The proposed method also sounds novel: to train the predictor by minimizing the distribution distance between the network's feature space and the label distribution.

3. The authors provide good ablation studies.

---

> ### Author Response · Authors · 2022-08-02
> **Response to Reviewer tD8p**
>
> We thank the reviewer for the insightful and positive comments.
>
> > Eq. 9, the scheduler for the Subspace Partition, sounds empirical. The authors may consider down-play its contribution, or compare Eq. 9 with other possible schedulers to show its importance.
>
> We thank the reviewer for this constructive suggestion. The core idea is to gradually increase the number of subspaces, and the specific formulation of Eq. (9) is indeed less important. We have changed the current Eq. (9) to $C_e = Sche(e;E,K)$ in the revised version. For the method proposed in this paper, we empirically find from Table 3 that the best results are obtained when the scheduler is set to $C_e = K - \lfloor cos( \frac{\pi}{2E} e)*K \rfloor$. More details can be checked from the revised version.
>
> > I can understand the importance of the Assistant Architecture Space. However, is there any possible principles in designing it? Otherwise it will hurt the generality of the proposed method.
>
> Thanks for the constructive comments. In Line 233 of our manuscript, we briefly introduce two principles for designing the assistant space, which are required in terms of size and characteristics. First, the size of the assistant space needs to be between the source space and the target space. Second, the assistant space needs to contain the characteristics of the source space and the target space (since both NAS-Bench and DARTS spaces are cell-based spaces, this is no point we need to pay much attention to when designing the assistant space in this paper). The search space designed can act as a bridge between the source space and the target space, which in turn helps the predictor to perform a smoother transfer. Since both of the above principles are relatively general, they can be applied to different search spaces generally.
>
> > d_H in Eq. (10) is not defined or discussed, or did I miss anything? I suppose it is a kind of distance measure between two domains?
>
> $d_H$ is the distance between two domains. To be specific, $d_H$ in Eq. (10) is the upper bound of the $\mathcal{A}$-distance [1]. We have included the corresponding introduction in Lemma 1 of the revised version.
>
> > I think Thm 2 is too general to be helpful to the main purpose of this (NAS) work. As far as I understand, the most insight from Thm. 2 is that larger m and m’ would give us a lower error bound. But this conclusion is very general and is not leveraged in this work (thus sounds less important).
>
> Theorem 2 accounts for the upper bound of the cross-domain predictor's error, which aims to establish a relationship between the two domains. Actually, Theorem 2 does not focus on $m$ and $m^\prime$. It shows that the expected error on the target domain $\epsilon_T(h)$ is bound by two terms, the empirical validation error in source domain $\hat{\epsilon}_{S, valid}(h)$ and MMD $2d_k(\widetilde{\mathcal{D}}_S,\widetilde{\mathcal{D}}_T)$. The other terms on the right side of the Eq. (11) inequality are generally regarded as constants because the number of samples $m$ and $m^\prime$ are determined based on the size of the particular dataset.
>
> Theorem 2 actually has a close relationship with our cross-domain predictor. As shown in Eq. (4), the objective function for optimization in this work contains two terms. For the first term $\frac{1}{n^s}\sum_{n=1}^{n^s} \mathcal{L}(P(W, \textbf{x}_{n}^s), y_{n}^s)$, we choose MSE as $\mathcal{L}$ to optimize the absolute error of the predictor on the training dataset and validate it on the validation dataset. In this way, the first term in Theorem 2, *i.e.*, the empirical validation error in source domain $\hat{\epsilon}_{S, valid}(h)$, is optimized. As for the second term $\theta d(\widetilde{\mathcal{D}}_S,\widetilde{\mathcal{D}}_T)$ in Eq (4), our work chooses MMD $\hat{d}_k(\widetilde{\mathcal{D}}_S,\widetilde{\mathcal{D}}_T)$ for the constraint. Therefore, the first two terms on the right side of Theorem 2 inequality are both optimized in this work. Hence, the expected error on the target domain $\epsilon_T(h)$ is small in practice, and the prediction performance of the predictor on the target domain is theoretically guaranteed.
>
> > I think in general the target problem and the method of this work is novel. However, some designs in the method (Eq. 9, and the assistant space) may be empirical and may hurt its practical use.
>
> Thanks for the approval. The main idea of Eq. (9) is to gradually increase the number of subspaces, and we have modified Eq. (9) to a more general form. This work provides a new perspective to predictor-assisted NAS, which may inspire the community. We will continue researching to improve the universality and practicality in the future.
>
> ------
> [1] Kifer, Daniel, Shai Ben-David, and Johannes Gehrke. "Detecting change in data streams." VLDB. Vol. 4. 2004.

---

> > ### Comment · Reviewer_tD8p · 2022-08-07
> > **Thanks for the response**
> >
> > 1. “both NAS-Bench and DARTS spaces are cell-based spaces” making this statement still requires human experts (especially experts in NAS). This indicates that designing the assistant space is not a trivial task.
> >
> > 2. For Eq. 11, I understand that the first two terms are optimized in the proposed algorithm. However, showing that one needs to optimize these two terms in order to achieve low error on target space is a kind of trivial conclusion. In other words, Thm 2 did not tell us how to further improve that: Shall we put more budget into collecting more samples in the source domain or the target domain? To put it another way, even without Thm 2, it is still easy to believe that one should optimize the source error and source-target distance.
> >
> > However, this is still a good paper and I would like to keep my score.

---

> > > ### Author Response · Authors · 2022-08-07
> > > **Thanks for the comments**
> > >
> > > We are grateful for the further comments and thanks very much for the approval.

---

### Official Review · Reviewer_Ywk3 · 2022-07-11

**Rating:** 7
**Confidence:** 4
**Soundness:** 3 good
**Presentation:** 3 good
**Contribution:** 3 good

**Summary:**

This paper develops a cross-domain predictor, called CDP, which does not require training neural architectures in the particular search space. It leverages the existing NAS benchmark datasets and adopts domain adaptation to adapt from the source space of NAS benchmark to the target space. Therefore, it does not require to train architectures in the target space and largely reduces the cost.

**Questions:**

1. How does CDP perform if the two domains are for different tasks? For example, from image classification to NLP task?
2. How is the kernel function and mapping function designed?



**Ethics Review Area:**

["I don’t know"]

**Strengths And Weaknesses:**

Strengths:
1. The motivation is clear to leverage the existing NAS benchmarks for training-free NAS.
2. The proposed method is sound with proof.
3. Improved experimental performances against other works on ImageNet and CIFAR-10.

Weaknesses:
1. The method relies on existing NAS benchmarks. Therefore, it is hard to apply to some domain/tasks that do not have collection of architecture-performance pairs.
2. The design of assistant space could be impractical for other domains/complex spaces. The NAS-Bench space and DARTS space are still simple.

---

> ### Author Response · Authors · 2022-08-02
> **Response to Reviewer Ywk3**
>
> We would like to thank the reviewer for carefully reading our submission and providing many insightful comments.
>
> > The method relies on existing NAS benchmarks. Therefore, it is hard to apply to some domain/tasks that do not have collection of architecture-performance pairs.
>
> Thanks for the constructive comments. Neural networks that can well handle similar tasks often have some common features that can still be used to identify good network architectures, even in different search spaces. In some tasks of computer vision, the architectures are similar in some cases, and well-performing architectures have commonality and consistency. For example, ResNet is often used as a backbone for detection and segmentation tasks, so there are features in these architectures that can be exploited by predictors. Thus the searched architecture has the potential to be applied to more tasks and domains.
>
> > The design of assistant space could be impractical for other domains/complex spaces. The NAS-Bench space and DARTS space are still simple.
>
> An assistant space can be designed following two principles. As discussed in Lines 233-236 of the submitted manuscript, the first principle is that the size of the assistant space needs to be between the source space and the target space, while the second principle is that the assistant space needs to contain as many unique features of both spaces as possible (since both NAS-Bench and DARTS spaces are cell-based spaces, this is not a point we need to pay much attention to when designing the assistant space). Furthermore, the motivation for these two design principles is that the designed assistant space should serve as a bridge between the source space and the target space, and make the transfer smoother. We believe that these two principles have the potential to work well on more complex spaces, and we will explore this interesting problem in the future.
>
> > How does CDP perform if the two domains are for different tasks? For example, from image classification to NLP task?
>
> The architectures for CV and NLP tasks indeed have a large difference (*e.g.*, CNN for computer vision tasks, RNN and transformer for NLP tasks). It is a large challenge to transfer architectures between them directly. Actually, even the general domain adaptation algorithms (*e.g.*, CKB[1] and DSAN[2]) only focus on similar domains, such as Office-Home[3], Image-CLEF-DA[4].
>
> Though it is a challenge, we have some ideas about it. In practice, the key to solve this issue is to find some features of the neural networks that are more general than the architectural information. When we find common features between CV networks and NLP networks (possibly some gradient features in training), we can easily then apply cross-domain predictors to the two tasks.
>
> > How is the kernel function and mapping function designed?
>
> We used the Gaussian kernel function, which is investigated by the ablation study in the supplementary material (Section E, More Ablation Studies). Specifically, we compare different kernel functions containing the Rational Quadratic kernel, Laplace kernel, and Gaussian kernel. Gaussian kernel empirically produces the best performance.
>
> ------
> [1] Luo, You-Wei, and Chuan-Xian Ren. "Conditional bures metric for domain adaptation." Proceedings of the IEEE/CVF Conference on Computer Vision and Pattern Recognition. 2021.
>
> [2] Zhu, Yongchun, et al. "Deep subdomain adaptation network for image classification." IEEE transactions on neural networks and learning systems. 2020.
>
> [3] Venkateswara, Hemanth, et al. "Deep hashing network for unsupervised domain adaptation." Proceedings of the IEEE conference on computer vision and pattern recognition. 2017.
>
> [4] Caputo, Barbara, et al. "ImageCLEF 2014: Overview and analysis of the results." International Conference of the Cross-Language Evaluation Forum for European Languages. Springer, Cham, 2014.

---

### Official Review · Reviewer_jvHq · 2022-07-11

**Rating:** 6
**Confidence:** 2
**Soundness:** 3 good
**Presentation:** 3 good
**Contribution:** 3 good

**Summary:**

Recent advances in Neural Architecture Search (NAS) investigate the neural predictor, which directly estimates the performance of neural architectures, to alleviate the costs of performance evaluation. However, the neural predictor still requires a set of annotated architectures for training. To alleviate this issue, this paper proposes a cross-domain adaptation strategy (CDP; Cross-Domain Predictor), which mitigates such training costs for annotation by utilizing existing labeled datasets.

**Questions:**

- How does this paper measure GPU hours? While the seminal work of DARTS [Liu et al., 2019] stated that “All of our experiments were performed using NVIDIA GTX 1080Ti GPUs.”, I cannot find such statements in this paper.
- Using Kendall’s Tau (KTau; Sen, 1968), in Section 4.3, the paper demonstrates the improvements from the proposed components for CDP (i.e., cosine scheduling, medium-sized assistant space, and LMMD) in terms of the quality of neural predictors. However, there are no comparisons with existing methods for training neural predictors, which utilize annotated datasets for the target space (e.g., Wen et al., 2020). It will be nice to check whether the proposed method produces high-quality neural predictors comparable to the previous approaches without such annotations.
---
- [Liu et al., 2019] DARTS: Differentiable Architecture Search. 2019.
- [Sen, 1968] Estimates of the Regression Coefficient Based on Kendall's Tau. 1968.
- [Wen et al., 2020] Neural Predictor for Neural Architecture Search. 2020.

**Limitations:**

This paper did not include both limitations and societal impacts of the work.

**Strengths And Weaknesses:**

Although this paper borrows existing concepts from domain adaptation, it does not hurt the significance of the work; I believe that converting key ideas across sub-fields (in this case, from domain adaptation into neural architecture search) is still valuable for academia.

Overall, the paper is well-written; it clearly presents the existing problems for training neural predictors and how the proposed method alleviates that. Applying domain adaptation methodologies to utilize existing labeled architecture space for another space seems reasonable. Moreover, NAS-specific improvements make the paper have its own contribution. I especially like the ablation study in Section 4.3, which measures the quality of the neural predictor for various settings of the proposed cross-domain adaptation strategies.

*Caveat: For the other experimental results in the paper (including Sections 4.1 and 4.2), I would defer to the other reviewers; since I am not familiar with NAS experiments.

---

> ### Author Response · Authors · 2022-08-02
> **Response to Reviewer jvHq**
>
> Thanks for the constructive comments.
>
> >How does this paper measure GPU hours? While the seminal work of DARTS [Liu et al., 2019] stated that “All of our experiments were performed using NVIDIA GTX 1080Ti GPUs.”, I cannot find such statements in this paper.
>
> In the experiment, we used NVIDIA RTX 2080Ti GPUs, and the search is done after running 0.1 GPU Days. Thanks for the suggestion, we have included the corresponding description in the revised version.
>
> In addition, we have also performed our method on a 1080Ti GPU, which required running 0.16 GPU Days. This search cost is still much better than the 4 GPU Days required by DARTS. Furthermore, it should be noted in Table 1 that the PC-DARTS method whose search cost can compete with our proposed CDP used the GPU of Tesla V100. Because both the performance and memory size of V100 are much higher than 2080Ti, this further proves the superiority of the search cost required by our proposed CDP method in NAS (Table 1 and Table 2).
>
> >Using Kendall’s Tau (KTau; Sen, 1968), in Section 4.3, the paper demonstrates the improvements from the proposed components for CDP (i.e., cosine scheduling, medium-sized assistant space, and LMMD) in terms of the quality of neural predictors. However, there are no comparisons with existing methods for training neural predictors, which utilize annotated datasets for the target space (e.g., Wen et al., 2020). It will be nice to check whether the proposed method produces high-quality neural predictors comparable to the previous approaches without such annotations.
>
> Thanks for the suggestion, we have added a comparison with traditional predictors on the KTau metric. We do our best to train 50 architectures in the DARTS space in such a limited time to construct a training dataset. After training the Neural Predictor (Wen et al., 2020) with the training dataset, its KTau on the shallow DARTS dataset is 0.1305 (while our proposed method is 0.5306). We have analyzed the reason for its inaccurate predictions: the search space of DARTS is too large ($10^{18}$) and we only sampled 50 architectures, so the predictor can be not sufficiently trained.
>
> In addition, as shown in Table 1, we have also compared the search results with many predictor-based methods, such as neural predictor and SemiNAS, and we can search for well-performed neural architectures with little cost. This again confirms the effectiveness of the proposed CDP method.

---

> > ### Comment · Reviewer_jvHq · 2022-08-09
> > **Thanks for the response**
> >
> > Thank you for your response. It is somewhat surprising that it is not necessary to measure GPU Days on the same device for benchmarking (it would be nice to specify the machine for each baseline, but if the convention is so, keep going). Although my concern on further comparison with traditional predictors is not yet satisfactorily resolved, I agree with the authors that the main results (i.e., Table 1) verify the effectiveness of the proposed method enough. I raise my score to 6 after the authors' rebuttal.

---

> > > ### Author Response · Authors · 2022-08-09
> > > **Thanks for the comments**
> > >
> > > Thanks for the feedback! We are grateful for the constructive comments and support.

---

### Official Review · Reviewer_KWRx · 2022-07-12

**Rating:** 5
**Confidence:** 4
**Soundness:** 3 good
**Presentation:** 3 good
**Contribution:** 3 good

**Summary:**

The paper set out to solve the domain discrepancy issue between source architecture space and the target space in Neural Architecture Predictors. The paper proposed a progressive subspace adaptation strategy to train a Cross-Domain Predictor.

**Questions:**

(1) I'm curious how would the performance be, if we only include NAS-Bench-101 or NAS-Bench-201 as source domain?
(2) In Table 1/2, would that be unfair that other methods do not have any transferred knowledge, while CDP has prior knowledge from NAS-Bench with the training time of NAS-Bench not included.

**Limitations:**

(a) Some of the up-to-date methods are not compared as stated in the Weakness part.
(b) Some additional ablation is needed as stated in Questions.

**Strengths And Weaknesses:**

Strengths:
(1) The proposed CDP method is simple and intuitive, this idea of transfering knowledge from NASBenches and leveraging LMMD loss from transfer learning/domain adaption sounds novel to me.
(2) The performance of transfer learning in NAS is impressive to me, though the search cost comparison is a bit misleading as stated in the questions.
(3) The paper is overall well-written and well-presented.

Weakness:
(1) The 0.5306 KTau in Table 5 is not impressive, the author should consider reporting the TopK% (or Top-50/100 depending on search space size) Kendall-Tau correlation of predictor, since NAS only output the BEST predicted architecture, that's where different predictor-based NAS methods would make a real difference.
(2) The comparison methods (NAO, Neural Predictor, SemiNAS) in Table 1 seem to be outdated, a comparison with more recent Predictor-based NAS methods [1][2] should also be included:
[1] BRP-NAS: Prediction-based NAS using GCNs, NeurIPS 2020
[2] Stronger NAS with Weaker Predictors. NeurIPS 2021

---

> ### Author Response · Authors · 2022-08-02
> **Response to Reviewer KWRx**
>
> We sincerely thank the anonymous reviewer for the support and constructive comments.
>
> > The 0.5306 KTau in Table 5 is not impressive, the author should consider reporting the TopK% (or Top-50/100 depending on search space size) Kendall-Tau correlation of predictor, since NAS only output the BEST predicted architecture, that's where different predictor-based NAS methods would make a real difference.
>
> We concur with the suggestion and have investigated the Kendall-Tau correlation upon the Top-5 architectures with the results shown below. As can be seen, our method still achieves higher precision than the competing methods, indicating its stronger ability to recognize the best architectures.
>
> |Approach |  Whole  | Top-5  |
> |-------- | ------  | ------ |
> |DANN     |  0.4686 | 0.4    |
> |CORAL    |  0.4306 | 0.4    |
> |Ours     |  0.5306 | 0.6    |
>
> > The comparison methods (NAO, Neural Predictor, SemiNAS) in Table 1 seem to be outdated, a comparison with more recent Predictor-based NAS methods [1][2] should also be included: [1] BRP-NAS: Prediction-based NAS using GCNs, NeurIPS 2020 [2] Stronger NAS with Weaker Predictors. NeurIPS 2021
>
> We further compare our method with these nice works. BRP-NAS[1] reports the search results on CIFAR-10, and WeakNAS[2] reports the results on ImageNet. Thus we add the comparison with both methods into Table 1 and Table 2 of the revised manuscript, respectively. As can be observed, in the same search space, our method can find architectures with higher accuracies than BRP-NAS and WeakNAS. For example, our method used 0.1 GPU Days to find an architecture with 97.37% $\pm$ 0.08% accuracy on CIFAR-10, which is more efficient than BRP-NAS (6 GPU Days and 97.29% $\pm$ 0.07% accuracy). For WeakNAS, it achieved the Top-1 accuracy of 76.5% on ImageNet, which is inferior to our method with an accuracy of 76.9%. Compared with them, our method still has a large superiority in search precision and efficiency. A more detailed comparison is shown in Tables 1 and 2 of the revised version.
>
> >  I'm curious how would the performance be, if we only include NAS-Bench-101 or NAS-Bench-201 as source domain?
>
> We have conducted experiments to investigate the impact of the dataset in the source domain. When using NAS-Bench-101 which contains 423K architecture as the source dataset, the predictor can achieve the KTau value of 0.4249. While only using NAS-Bench-201, which is a small dataset containing only 15K architectures, a lower KTau of 0.1574 is obtained. When the two datasets are used together, the predictor’s precision can be significantly improved with the KTau of *0.5306*.
>
> > In Table 1/2, would that be unfair that other methods do not have any transferred knowledge, while CDP has prior knowledge from NAS-Bench with the training time of NAS-Bench not included.
>
> The other methods and our method in this paper are all developed for the same goal to improve the search process, which is a critical issue of current NAS algorithms. The source data we used are benchmark datasets that are publicly available and also are cheap for utilization. Our motivation in this paper is to exploit this cheap data from a new perspective to address the critical issue. On the other hand, we are also the first work to take advantage of these datasets, which means that previous approaches could not utilize these cheap resources.

---

### Meta-Review · Area_Chair_W4JG · 2022-08-28

**Recommendation:** Accept
**Confidence:** Certain

**Metareview:**

This paper introduces ideas from domain adaptation to improve NAS: leveraging leverage existing NAS Bench to predict out-of-domain architecture’s performance. This is an important question that has been overlooked, and the authors propose to learn a predictor via closing the domain feature gap between source and target architecture space. Both the problem setting and the proposed methods are novel. The authors also present good ablation studies, and the rebuttal was able to address a few clarification questions. After the rebuttal, all reviewers seem to be positive about this work, and the AC sides with them.

**Award:**

No

---

### Decision · Program_Chairs · 2022-09-14

Accept